# Studies on the Properties of the Sporulation Specific Protein Dit1 and Its Product Formyl Tyrosine

**DOI:** 10.3390/jof6020077

**Published:** 2020-06-03

**Authors:** Mostafa Basiony, Yan Yang, Guoyu Liu, Xiao-Dong Gao, Hideki Nakanishi

**Affiliations:** Key Laboratory of Carbohydrate Chemistry and Biotechnology, Ministry of Education, School of Biotechnology, Jiangnan University, Wuxi 214122, China; mostafabasiony@ymail.com (M.B.); yangyang201298@163.com (Y.Y.); lgy17091037820@163.com (G.L.)

**Keywords:** Dit1, formyl tyrosine, dityrosine layer, *Saccharomyces cerevisiae*, spore wall

## Abstract

The dityrosine layer is a unique structure present in the spore wall of the budding yeast *Saccharomyces cerevisiae*. The primary constituent of this layer is bisformyl dityrosine. A sporulation-specific protein, Dit1 is localized in the spore cytosol and produces a precursor of bisformyl dityrosine. Although Dit1 is similar to isocyanide synthases, the loss of Dit1 is not rescued by heterologous expression of the *Pseudomonas aeruginosa* isocyanide synthase, PvcA, indicating that Dit1 does not mediate isocyanidation. The product of Dit1 is most likely formyl tyrosine. Dit1 can produce its product when it is expressed in vegetative cells; however, formyl tyrosine was not detected in the crude cell lysate. We reasoned that formyl tyrosine is unstable and reacts with some molecule to form formyl tyrosine-containing molecules in the cell lysate. In support of this hypothesis, formyl tyrosine was detected when the lysate was hydrolyzed with a mild acid. The same property was also found for bisformyl dityrosine. Bisformyl dityrosine molecules assemble to form the dityrosine layer by an unknown mechanism. Given that bisformyl dityrosine can be released from the spore wall by mild hydrolysis, the process of formyl tyrosine-containing molecule formation may resemble the assembly of the dityrosine layer.

## 1. Introduction

Under starvation conditions, the diploid cells of the budding yeast *Saccharomyces cerevisiae* undergo meiosis and form spores [1]. Spores are formed inside of the mother cells, where each haploid nucleus generated via meiosis is enclosed by the spore plasma membrane and spore wall [2]. The spore wall consists of four layers, which are (from inside to outside) mannan, glucan, chitosan and dityrosine [2]. During sporulation, these structures are synthesized de novo in a sequential manner from the inner layer [3]. The outermost layer, the dityrosine layer, is a unique structure of the *S. cerevisiae* spore wall. Although the dityrosine layer is dispensable for making viable spores, it confers resistance to spores against environmental stresses [4].

It is believed that the dityrosine layer is formed by the assembly of bisformyl dityrosine molecules [5]. Bisformyl dityrosine is formed in the spore cytosol and transported to the nascent spore wall [4,6,7]. Two sporulation-specific proteins, Dit1 and Dit2, are known to be involved in the synthesis of bisformyl dityrosine [4,6]. Dit1 is believed to be an enzyme that mediates the *N*-formylation of tyrosine to produce formyl tyrosine [5,6]. However, since the enzymatic activity of Dit1 has not been well characterized, it remains possible that the product of Dit1 is not formyl tyrosine. Moreover, it is notable that the primary structure of Dit1 is similar to bacterial isocyanide synthases [8]. In *Pseudomonas aeruginosa*, the isocyanide synthase PvcA is known to mediate isocyanide functionalized tyrosine as a precursor to producing paerucumarin [9]. Because the isocyanide group is unstable, paerucumarin can be converted to its *N*-formyl adduct pseudoverdine by the addition of water [9,10].

Dit2 is a P450 family protein that most likely mediates the crosslinking of formyl tyrosine molecules to generate bisformyl dityrosine [4,11]. Previously, we performed an in vitro Dit2 assay [11]. In the assay, Dit2 can crosslink formyl tyrosine, but not tyrosine, suggesting that formyl tyrosine is specifically recognized as the substrate [11]. It should be noted that bisformyl dityrosine was not detected in the reaction mixture in our assay, although studies conducted by another group showed that bisformyl dityrosine was detected in yeast lysates [5,6]. Thus, in our assay, the activity of Dit2 was monitored by the detection of dityrosine in the reaction mixture after treatment with a strong acid. One possible explanation for the failure to detect bisformyl dityrosine in our assay is that bisformyl dityrosine molecules produced under our experimental conditions might react with other molecules in the reaction mixture [11]. The hypothetical molecule was referred to as a bisformyl dityrosine-containing molecule. Bisformyl dityrosine incorporated in the bisformyl dityrosine-containing molecules could be released by strong acid hydrolysis. Since the formyl group is removed by strong acid hydrolysis, dityrosine was detected in the hydrolyzed reaction mixture [11].

In the present study, we report the properties of Dit1 and its product, formyl tyrosine. The primary obstacle for the analysis of Dit1 activity is that formyl tyrosine was not detected in yeast lysates. Similar to bisformyl dityrosine, formyl tyrosine likely forms formyl tyrosine-containing molecules. We found that formyl tyrosine can be liberated from formyl tyrosine-containing molecules by mild hydrolysis. The results presented here would be useful for further analysis of Dit1 and the dityrosine layer.

## 2. Materials and Methods

### 2.1. Yeast Strains

The yeast strains and primers used in this study are listed in Table 1 and Table 2, respectively. To integrate GFP at the 3′-end of the chromosomal *DIT1* genes, a DNA fragment for the integration was amplified by PCR using pFA6a-GFP (S65T)-HIS3MX6 [12] as a template and DIT1-GFP-F-INT and DIT1-GFP-R-INT primers. The PCR fragment was integrated into AN117-4B and AN117-16D [13], and the resulting haploid strains were crossed to generate diploid cells (HI391). The strain AN120 [13] was used for experiments using spores (Figure 1 and Figure 2). For other experiments, the strain YPH499 [12] was used. 

### 2.2. Plasmids

The plasmids used in this study are listed in Table 3. The *DIT1* expression plasmid pRS424GAL1-DIT1-GFP was constructed as follows: first, GFP gene was amplified by PCR using the primers HP128 and HP129 and pSGFP2-C1 [15] as a template. The GFP fragment was subcloned into *Hin*dIII/*Xho*I sites of pRS316. Then, the GFP gene was digested out of the resulting plasmid with *Eco*RI and *Xho*I, and cloned into similarly digested pRS424GAL1 [16]. The resulting plasmid was named pRS424GAL1-GFP. The *DIT1* gene (without stop codon) was amplified by PCR using the primers HXO683 and ECOR1-DIT1-r, and yeast genomic DNA was used as a template. The PCR fragments were digested with *Spe*I and *Eco*RI and ligated into similarly digested pRS424GAL1-GFP. For pRS426GAL1-DIT2-FLAG construction, the *DIT2-FLAG* gene was amplified by PCR using the HXO685 and DIT2-FLAG-r as primers and yeast genomic DNA as a template. The PCR fragment was cloned into the *Spe*I/*Bam*HI sites of pRS426GAL1 [11]. The *pvcA* gene was expressed under the control of the *TEF2* promoter using pRS316TEF-PvcA. To construct this, the *pvcA* gene synthesized by GENEWIZ (Suzhou, China) was cloned into the *Spe*I/*Xho*I sites of pRS316TEF [17]. For the construction of the pRS424GAL1pr-Dit1-HA plasmid, the Dit1-HA fragment was amplified using HXO683 and DIT1-HA-r as primers and yeast genomic DNA as a template. The PCR fragment was digested with *Spe*I and *Eco*RI and ligated into similarly digested pRS424GAL1.

### 2.3. Yeast Culture and Sporulation 

Yeast culture media were prepared as previously described [18]. Yeast sporulation was performed as described previously [14] with some modifications. In brief, yeast cells derived from a single colony were cultured overnight in 5 mL of YPAD or SD media supplemented with the appropriate amino acids. One milliliter of the culture was added to 200 mL of YPAcetate medium for 24 h. Next, the cells were harvested, washed twice with distilled water and resuspended in 100 mL of sporulation medium. The AN120 strain culture was incubated at 30 °C for 24 h (Figure 1A), and the HI391 strain was incubated for 12 h (Figure 2).

For ectopic expression of *DIT1* and *DIT2* in vegetative cells, yeast cells carrying pRS424GAL1-DIT1-GFP or pRS426GAL1-DIT2-FLAG were cultured in 50 mL of SD glucose medium supplemented with the appropriate amino acids overnight. Then, cells were collected by centrifugation and washed twice with distilled water. After washing, the cells were resuspended in 30 mL of SD galactose containing medium and cultured for 8 h. All cells were cultured at 30 °C.

### 2.4. In vitro Assays for Dityrosine 

The Dit2 activity assay was performed as described before [11] with some modifications. Fifty-two milligrams of vegetative cells harboring pRS426GAL1-DIT2-FLAG were permeabilized with 50 µL of 0.5% Triton X-100, or lysed using glass beads (0.5 mm diameter; Sigma-Aldrich, Shanghai, China) in 50 µL of cold PBS buffer (137 mM NaCl, 2.7 mM KCl, 10 mM Na_2_HPO_4_, 2 mM KH_2_PO_4_) supplemented with proteinase inhibitor cocktail (MedChemExpress, NJ, USA). The lysates were centrifuged for 5 min at 3000× *g* to remove the cell debris and glass beads. Then, 50 µL of 2 mg/mL *N*-formyl tyrosine (Sigma-Aldrich) was added, and the reaction mixture was incubated at 30 °C for 4 h and centrifuged at 21,400× *g* for 5 min and the supernatant was retained for further analysis.

For the Dit1 activity assay, 104 mg of vegetative cells harboring pRS424GAL1-DIT1-GFP were permeabilized with 50 µL of 0.5% Triton X-100, or lysed using glass beads in 50 µL of cold PBS buffer (137 mM NaCl, 2.7 mM KCl, 10 mM Na_2_HPO_4_, 2 mM KH_2_PO_4_) supplemented with proteinase inhibitor cocktail. The lysates were centrifuged for 5 min at 3000× *g* to remove the cell debris and glass beads. Then, 50 µL of Dit2-expressing cell lysates were prepared as mentioned above, and 50 µL of 2 mg/mL tyrosine was added. The reaction mixture was incubated at 30 °C for 4 h, centrifuged at 21,400× *g* for 5 min, and the supernatant was retained for further analysis 

### 2.5. High Performance Liquid Chromatography (HPLC) Analysis

To prepare cell lysates, 52 mg of cells were suspended in 100 µL of cold PBS buffer supplemented with proteinase inhibitor cocktail and lysed with glass beads. Samples were centrifuged at 21,400× *g* for 20 min. The supernatant was treated as follows to prepare the HPLC samples. For the crude samples, the supernatant was filtered through 0.45 µm microfilters and subjected to HPLC analysis.

Complete acid hydrolysis was performed as described previously [14]. Two hundred microliters of 6 N HCl was added to 100 µL of the supernatant, and incubated at 95 °C for 5 h with an open lid. The dried hydrolysates were resuspended in 200 µL of distilled water, centrifuged for 5 min at 21,400× *g* and filtered through 0.45 µm microfilters.

Acid hydrolysis with trisodium citrate was performed as described previously [19] with some modifications. First, 4 µL of 1 M trisodium citrate was added to 196 µL of the supernatants, and the mixture was incubated at 100 °C for 1 h with a closed lid. The samples were centrifuged at 21,400× *g* for 5 min. The supernatant was transferred to new tubes and distilled water was added to dilute it to 200 µL. Samples were then filtered through 0.45 µm microfilters and subjected to HPLC analysis. 

HPLC analysis was performed as follows. The samples were analyzed with a Discovery C18 column (150 mm × 4.6 mm ID, 5 µm particles) (Sigma-Aldrich) using a Waters separation module e2695 HPLC system (Wexford, UK). Ten microliters of each sample was loaded. The column was developed with a gradient of CH_3_CN in 0.01 M trifluoroacetic acid (0%–50% CH_3_CN over 55 min). The flow rate was 1 mL/min. The fluorescence detector was set at 285 nm excitation and 425 nm emissions for dityrosine detection, and at 274 nm excitation and 303 nm emissions for tyrosine detection.

Dityrosine and *N, N*-bisformyl dityrosine were synthesized by the oxidation of tyrosine and *N*-formyl tyrosine, respectively, as previously described (18). In brief, 150 µL of 1 M Tris-HCl, 1.5 mL of 2 mg/mL tyrosine or *N-*formyl tyrosine (Sigma-Aldrich), 100 µL of 0.003% hydrogen peroxide and 500 µL of 1 mg/mL horseradish peroxidase (Sangon, shanghai, china) were mixed in and diluted up to 5 mL with water and incubated at 20 °C for 1 h.

### 2.6. Western Blot Analysis

Western blotting was performed as described previously [20] with some modifications. Vegetative cells or spores (52 mg) were lysed with glass beads in 100 µL of cold PBS buffer supplemented with a proteinase inhibitor cocktail and 1% NP-40 (Beyotime, Jiangsu, China). The cell lysates were centrifuged at 4 °C for 20 min at 21,400× *g*, and 50 µg of the supernatants were subjected to SDS-PAGE (5% stacking gel and 10% separating gel). The protein concentration was measured using a BCA protein assay kit (Beyotime). Mouse anti-GFP antibodies (1:3000) (Transgen Biotech, Beijing, China) and rabbit anti-FLAG antibodies (1:3000) (Sigma-Aldrich) were use as primary antibodies. Goat anti-mouse IgG-HRP (1:2000) (Transgen Biotech) and Goat anti-rabbit IgG-HRP (1:2000) (Transgen) were used as secondary antibodies. Bands were visualized by Clarity Western ECL Substrate (Bio-Rad, Shanghai, China), and images were obtained by using ImageQuant LAS4000 (GE Healthcare Bio-Science, Uppsala, Sweden).

### 2.7. Coimmunoprecipitation

Yeast cells and lysates were prepared as described in the previous section. Mouse anti-HA antibodies (1:200) (Transgen) were incubated with 200 µL of lysate for 1 h at 4 °C. Forty microliters of protein A + G beads slurry (Beyotime) was washed three times with PBS and centrifuged at 3000× *g* for 1 min. Then, the beads were resuspended in 40 µL of PBS, and incubated with the lysate-antibody mixture for 2 h at 4° C. The beads were collected at 3000 × *g* for 5 min and washed five times with PBS, resuspended in SDS-PAGE sample buffer, and subjected to Western blotting. 

### 2.8. Microscopy

Microscopy images were obtained using a Nikon Eclipse Ti-E inverted microscope equipped with a DS-Ri camera and NIS-Element AR software (Nikon, Tokyo, Japan). To stain spores with calcofluor white (CFW), 52 mg of spores were suspended in 200 µL of distilled water and mixed with 20 µL of 1 mg/mL CFW (Sigma-Aldrich). The mixture was incubated at 30 °C for 30 min, and then washed three times with sterile water. Cells were resuspended in 1 mL of distilled water, and then the fluorescence was examined by fluorescence microscopy. 

### 2.9. Statistics

The data presented are the mean ± SE of six independent samples. The statistical significance was calculated with Student’s t-test (two tail, paired) using Microsoft Excel Software (Version 16.0.4266.1001, Microsoft, Redmond, WA, USA). The differences between the analyzed samples was considered significant at *p* < 0.05.

## 3. Results

### 3.1. Dit1 Is not an Isocyanide Synthase

Dit1 is similar to isocyanide synthases [8]. The isocyanide group is unstable and can be easily hydrolyzed into a formyl group [9,21]. Thus, the product of Dit1 may be isocyanide-functionalized tyrosine. To assess this possibility, we examined whether an isocyanide synthase of *P. aeruginosa*, PvcA, can rescue the deficiency of the dityrosine layer formation in *dit1*Δ spores. Calcofluor white (CFW), which is a dye that stains chitosan, was used as an indicator to analyze the presence of the dityrosine layer. The spore wall contains chitosan but spores are not stained with CFW because the dityrosine layer prevents the binding of CFW to the chitosan layer in the spore wall [3]. In contrast, the chitosan layer is stained with CFW in *dit1*Δ spores. PvcA was expressed under the control of the constitutive *TEF2* promoter in *dit1Δ* cells. As shown in Figure 1A, *dit1Δ* spores expressing PvcA were stained with CFW, suggesting that PvcA cannot rescue the loss of Dit1. Dityrosine was not detected in *dit1∆* cells expressing PvcA (Figure 1B). Furthermore, we found that 20% of wild-type cells expressing PvcA were stained with CFW whereas 0.6% of cells harboring the empty vector were stained with CFW (Figure 1A). This result suggests that dityrosine formation is prevented by the activity of PvcA. To verify that the amount of dityrosine was decreased by expression of PvcA, dityrosine liberated from the spore wall was measured by HPLC. As shown in Figure 1C,D, the amount of dityrosine was decreased in spores expressing PvcA compared to those harboring the empty vector. Dityrosine molecules liberated from the spore wall exhibited doublet peaks because they are mixtures of the ll-and dl-forms (Figure 1C) [22]. These results demonstrate that Dit1 is not an isocyanide synthase.

### 3.2. Dit1 Expressed in Vegetative Cells Was Active but Formyl Tyrosine Was not Detected in the Lysate

To detect Dit1 optically and biochemically, a GFP tag was fused to the 3′-end of the chromosomal *DIT1* genes in the diploid strain. This strain was resistant to CFW staining (Appendix A), suggesting that the function of Dit1 was not interfered with by the addition of GFP at the C-terminus. Expression of the GFP fusion protein was verified by Western blotting. Since Dit1 is a sporulation-specific protein [4], the Dit1-GFP fusion protein was detected in the lysate of spores but not vegetative cells (Figure 2A). Dit1 does not contain either a predicted transmembrane domain or a signal peptide. Accordingly, Dit1-GFP was found in the cytosol of spores (Figure 2B). 

To further characterize Dit1, it was expressed in vegetative cells under the control of an inducible *GAL1* promoter (Figure 3B). Dit1-GFP expressed in vegetative cells was observed in the cytosol (Figure 3A). To analyze formyl tyrosine production in the Dit1-expressing cells, the cell lysate was subjected to HPLC analysis. Although a peak similar to formyl tyrosine was detected in the lysate of Dit1-GFP-expressing cells, the same peak was also detected in the control cell lysate (Figure 3C). Furthermore, the height of the peak was not significantly altered by the expression of Dit1-GFP (Figure 3C), suggesting that this peak is not formyl tyrosine. In a previous study, we showed that Dit2 expressed in vegetative cells was active and could crosslink formyl tyrosine [11]. Thus, to verify whether Dit1 was active in vegetative cells, Dit1-GFP and Dit2-FLAG were coexpressed in vegetative cells. Both Dit1-GFP and Dit2-FLAG were expressed under the control of the *GAL1* promoter (Figure 4A). As reported, previously, we could not detect bisformyl dityrosine under our experimental conditions [11]. Nevertheless, we detected dityrosine in the lysate of cells expressing Dit1-GFP and Dit2-FLAG after hydrolysis with HCl (Figure 4B). Dityrosine was not detected when either Dit1 or Dit2 was solely expressed (Figure 4B). Thus, Dit1-GFP is active in vegetative cells.

### 3.3. Dit1 Produces Formyl Tyrosine but It Is not Stable in Yeast Lysate

The absence of formyl tyrosine in the lysate of Dit1-GFP-expressing vegetative cells was further verified by an in vitro Dit2 assay. Dit2 expressed in vegetative cells retained its activity in permeabilized cells [11]. We also found that Dit2 was active in the cell lysate; when formyl tyrosine molecules were added to the cell lysate harboring Dit2, dityrosine was detected in the hydrolysate of the reaction mixture (Figure 5A). However, we could not detect dityrosine when the lysate of Dit1 expressing cells was incubated with the lysate of Dit2 expressing cells (Figure 5B). 

Then, we examined whether Dit1-GFP-expressing cells were permeabilized in the lysate of Dit2-expressing cells; therefore, Dit1-GFP-expressing cells were mildly lysed, and the lysate was immediately incubated with Dit2. Under this condition, we detected dityrosine in the hydrolysate of the reaction mixture (Figure 5B). Thus, we speculated that formyl tyrosine might be unstable and react immediately with other molecules in the lysate. 

To test this hypothesis, we sought a method to liberate formyl tyrosine from the formyl tyrosine-containing molecule. For this purpose, strong hydrolysis with HCl was not appropriate because tyrosine, which would be liberated by this treatment, was present in the yeast lysate (Figure 3C). A previous study showed that bisformyl dityrosine molecules can be released from the spore wall by trisodium citrate treatment [5]. We verified that bisformyl dityrosine could be detected by HPLC in the eluent of spores treated with trisodium citrate (Appendix A). In contrast to strong acid hydrolysis with HCl, trisodium citrate does not cause complete removal of the formyl group. Thus, we examined whether formyl tyrosine is liberated from formyl tyrosine-containing molecules by this mild hydrolysis. As shown in Figure 6A, we found a peak that appeared in the lysate of Dit1-GFP-expressing cells upon trisodium citrate treatment. This peak was not detected in the lysate of the cells harboring the empty vector (Figure 6A). A fraction from this peak was collected and hydrolyzed with HCl to verify that the peak fraction included formyl tyrosine. After the hydrolysis with HCl, a tyrosine peak was detected in the sample (Figure 6B). 

Next, we examined whether bisformyl dityrosine was also liberated by mild hydrolysis. When the lysate of Dit1-GFP and Dit2-FLAG coexpressing cells was hydrolyzed with trisodium citrate, bisformyl dityrosine peak was detected by HPLC (Figure 7A). We verified that dityrosine was detected when the collected bisformyl tyrosine peak was hydrolyzed with HCl (Figure 7B). These results indicate that formyl tyrosine moieties are not stable and that formyl tyrosine- and bisformyl dityrosine-containing molecules are formed in yeast lysates. 

## 4. Discussion

Although Dit1 is similar to isocyanide synthases, the product of Dit1 is not isocyanide-functionalized tyrosine. In wild-type cells, the amount of dityrosine was decreased with the expression of PvcA, probably because Dit1 and PvcA compete for the same substrate. The acceptor and donor substrates of PvcA are tyrosine and ribulose-5-phosphate, respectively [8]. Since the donor substrate for Dit1 remains unknown, it would be intriguing to test whether ribulose-5-phosphate is used as a substrate for Dit1.

The product of Dit1 is most likely formyl tyrosine. However, formyl tyrosine was not detected in the cell lysate. We found that formyl tyrosine can be detected in the lysate of Dit1-expressing cells after hydrolyzation with trisodium citrate. Our results collectively suggest that formyl tyrosine is unstable and is likely to form formyl tyrosine-containing molecules. The results of the in vitro Dit2 assay showed that the formyl tyrosine-containing molecules are not used as substrates for Dit2. Notably, dityrosine was detected in the reaction mixture when Dit1-expressing cells were permeabilized in the lysate of Dit2-expressing cells. A crosslinking reaction occurred under these conditions, possibly because the formyl tyrosine molecules generated in Dit1-expressing cells were immediately crosslinked by Dit2. It remains uncertain whether the formyl tyrosine-containing molecule is formed in the cytosol of live cells. If formyl tyrosine-containing molecules are generated in vivo, bisformyl dityrosine should be produced immediately from formyl tyrosine. However, Dit1 and Dit2 localize to different places, as Dit1 is a cytosolic protein while Dit2 is localized to the endoplasmic reticulum [11]. Furthermore, we performed a coimmunoprecipitation experiment but a physical interaction between Dit1 and Dit2 was not detected (Appendix A). These results suggest that the Dit1 and Dit2 reactions do not take place consecutively. Thus, it is quite possible that a formyl tyrosine-containing molecule is generated during the preparation of the cell lysate. In support of this possibility, previous reports showed that formyl tyrosine can be detected in yeast lysates [5,6]. Between this and the previous studies, the primary difference is that the lysates were deproteinized in the previous studies. Thus, proteins may be involved in the formation of formyl tyrosine-containing molecules.

Similar to formyl tyrosine, bisformyl dityrosine was not detected in the yeast lysate, likely because it formed a bisformyl dityrosine-containing molecule. In line with this hypothesis, bisformyl dityrosine can be detected when the cell lysate is hydrolyzed with a mild acid. In sporulating cells, bisformyl dityrosine molecules are assembled by an unknown mechanism to form the dityrosine layer. Since bisformyl dityrosine molecules are also released by mild acid hydrolysis from the spore wall [5], they may be incorporated into the spore wall via the same linkage that forms the bisformyl dityrosine-containing molecules. Thus, further analysis of the formyl tyrosine- and bisformyl dityrosine-containing molecules would provide an insight into the assembly mechanism of the dityrosine layer. The dityrosine layer is assembled on the chitosan layer and endows spores with unique properties [2]. Thus, formyl tyrosine and bisformyl dityrosine may be intriguing molecules from the point of view of material science.

## Figures and Tables

**Figure 1 jof-06-00077-f001:**
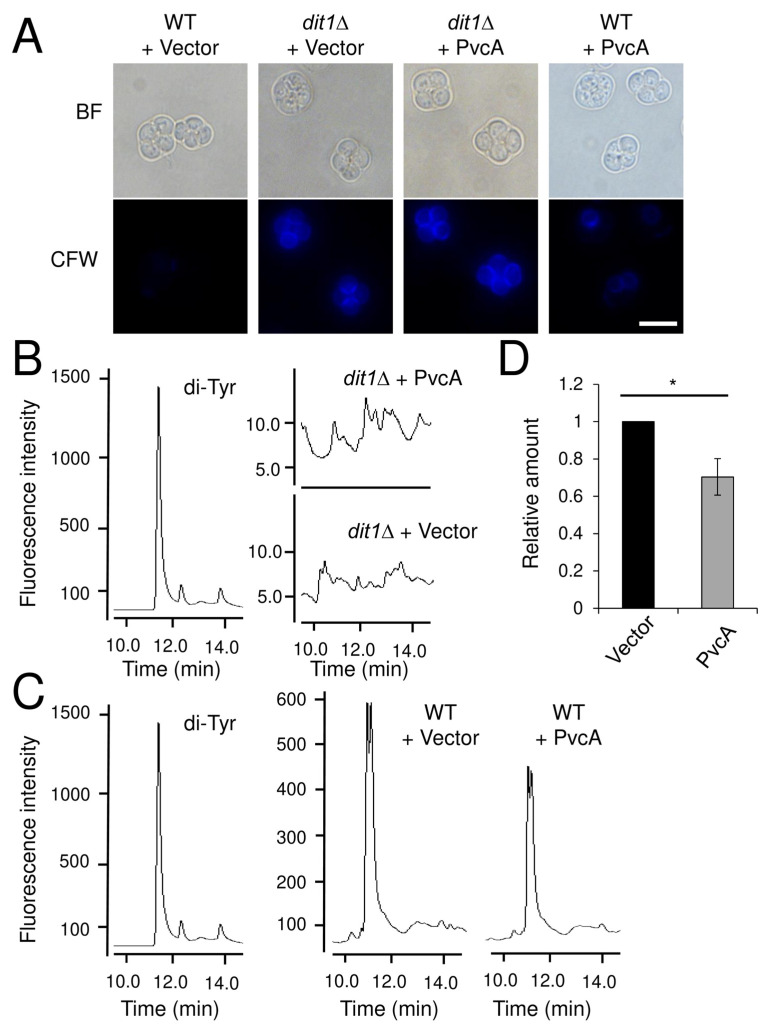
Heterologous expression of *P. aeruginosa* PvcA in yeast cells. (**A**) Wild-type (WT) or *dit1Δ* spores harboring pRS316TEF-PvcA (PvcA) or pRS316TEF (Vector), were stained with CFW and observed by bright field (BF) or florescence microscopy (CFW). Bar, 5 µm. (**B**) Lysates of *dit1Δ* spores harboring pRS316TEF-PvcA (PvcA) and pRS316TEF (Vector) were hydrolyzed with 6 N HCl and subjected to HPLC analysis. Dityrosine (di-Tyr) is shown as a control. (**C**) Lysates of wild-type spores harboring pRS316TEF-PvcA (PvcA) or pRS316TEF (Vector) were hydrolyzed with 6 N HCl and subjected to HPLC analysis. Dityrosine (di-Tyr) was assayed as a control. (**D**) Lysates of wild-type spores harboring pRS316TEF-PvcA (PvcA) or pRS316TEF (Vector) were hydrolyzed with 6 N HCl and subjected to HPLC analysis. The relative amount of dityrosine detected by HPLC is shown. The data presented are the mean ± SE of six independent samples. * *p* < 0.05.

**Figure 2 jof-06-00077-f002:**
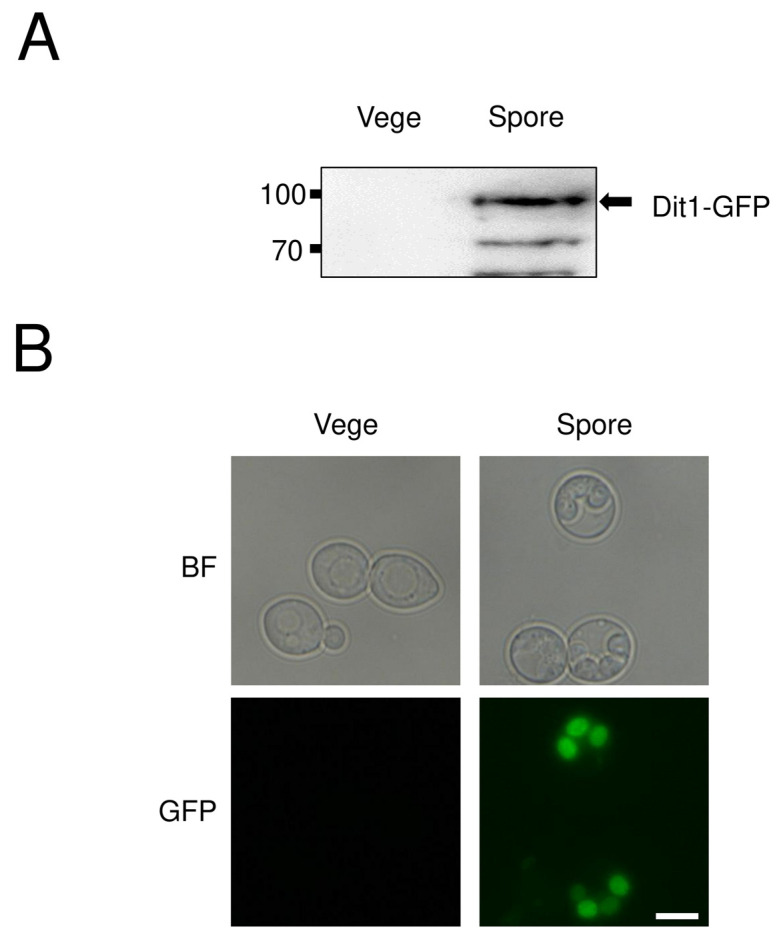
Analysis of Dit1 localization. (**A**) A diploid strain in which the chromosomal *DIT1* genes were fused to GFP. Vegetative growing cells (Vege) and spores of the *DIT1-GFP* harboring cells were lysed and subjected to Western blotting analysis using anti-GFP antibodies. (**B**) Vegetative growing cells (Vege) and spores of the *DIT1-GFP* harboring cells were observed by bright field (BF) and florescence microscopy (GFP). Bar, 5 µm.

**Figure 3 jof-06-00077-f003:**
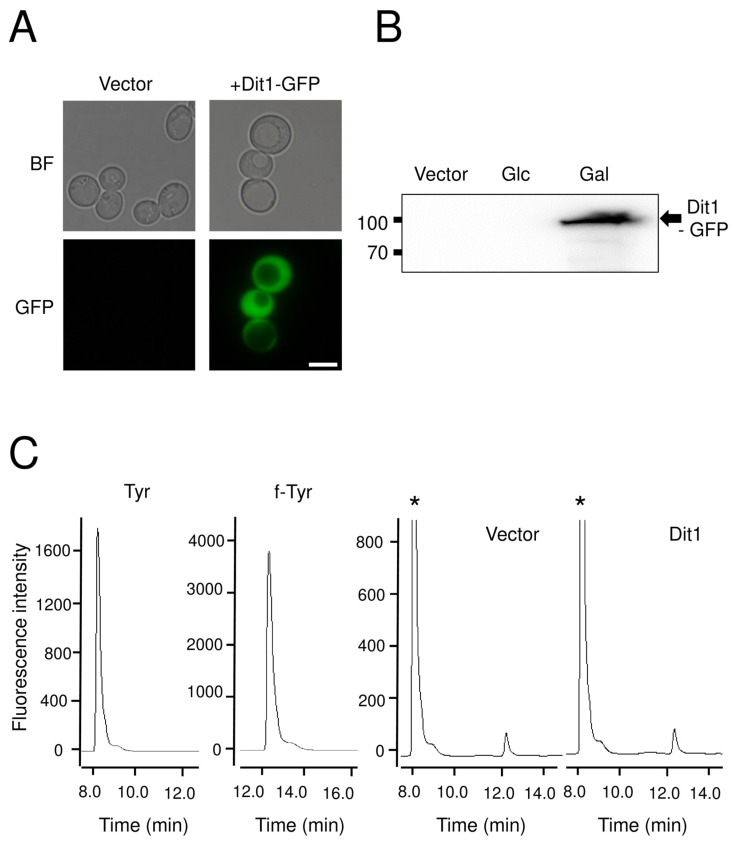
Ectopic expression of Dit1 in vegetative cells and analysis of formyl tyrosine production. (**A**) Wild-type vegetative cells harboring pRS424GAL1 (Vector) or pRS424GAL1-DIT1-GFP (+Dit1-GFP) were cultured in galactose-containing media and observed by bright field (BF) or florescence microscopy (GFP). Bar, 5 µm. (**B**) Lysates from wild-type vegetative cells harboring pRS424GAL1-DIT1-GFP cultured in glucose (Glc) or galactose (Gal) containing media were subjected to Western blotting analysis using anti-GFP antibodies. (**C**) Lysates of vegetative cells harboring pRS424GAL1 (Vector) or pRS424GAL1-DIT1-GFP (Dit1) were analyzed by HPLC. Stars indicate tyrosine peaks. Tyrosine (Tyr) and formyl tyrosine (f-Tyr) are shown as controls.

**Figure 4 jof-06-00077-f004:**
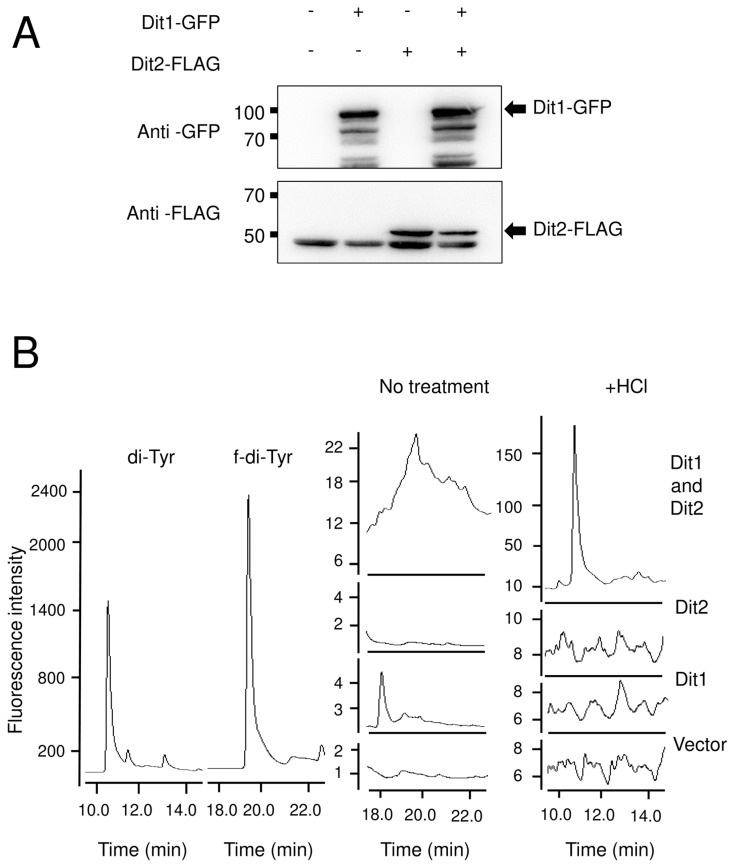
Dit1-GFP is active in vegetative cells. (**A**) Lysates from wild-type vegetative cells harboring pRS424GAL1-DIT1-GFP (Dit1-GFP) and/or pRS426GAL1-DIT2-FlAG (Dit2-FLAG) were subjected to Western blotting analysis using anti-GFP antibodies and anti-FLAG antibodies. (**B**) Lysates of vegetative cells harboring pRS424GAL1-DIT1-GFP (Dit1) and/or pRS426GAL1-DIT2-FlAG (Dit2) were treated with or without (no treatment) 6 N HCl and subjected to HPLC analysis. Dityrosine (di-Tyr) and formyl dityrosine (f-di-Tyr) are shown as controls.

**Figure 5 jof-06-00077-f005:**
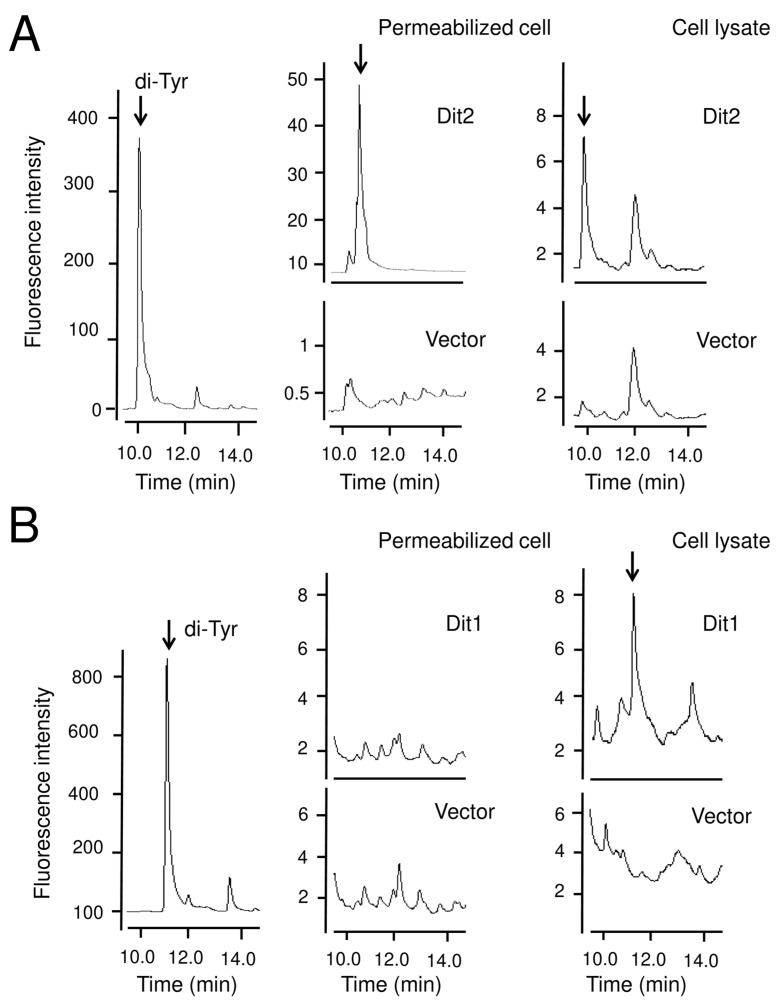
In vitro Dit2 assay to assess the presence of formyl tyrosine in Dit1-expressing cells. (**A**) Vegetative cells harboring pRS426GAL1-DIT2-FlAG (Dit2) or pRS426GAL1 (Vector) were permeabilized or lysed and incubated with formyl tyrosine (f-Tyr). The reaction mixtures were hydrolyzed with 6 N HCl and analyzed by HPLC. Di-tyrosine (di-Tyr) is shown as a control. Arrows indicate dityrosine peaks. (**B**) Lysates of vegetative cells harboring pRS424GAL1-DIT1-GFP (Dit1) were permeabilized or lysed and mixed with lysates of vegetative cells harboring pRS426GAL1-DIT2-FlAG (Dit2). The mixtures were incubated with tyrosine. The reaction mixtures were hydrolyzed with 6 N HCl and analyzed by HPLC. Dityrosine (di-Tyr) is shown as a control. Arrows indicate dityrosine peaks.

**Figure 6 jof-06-00077-f006:**
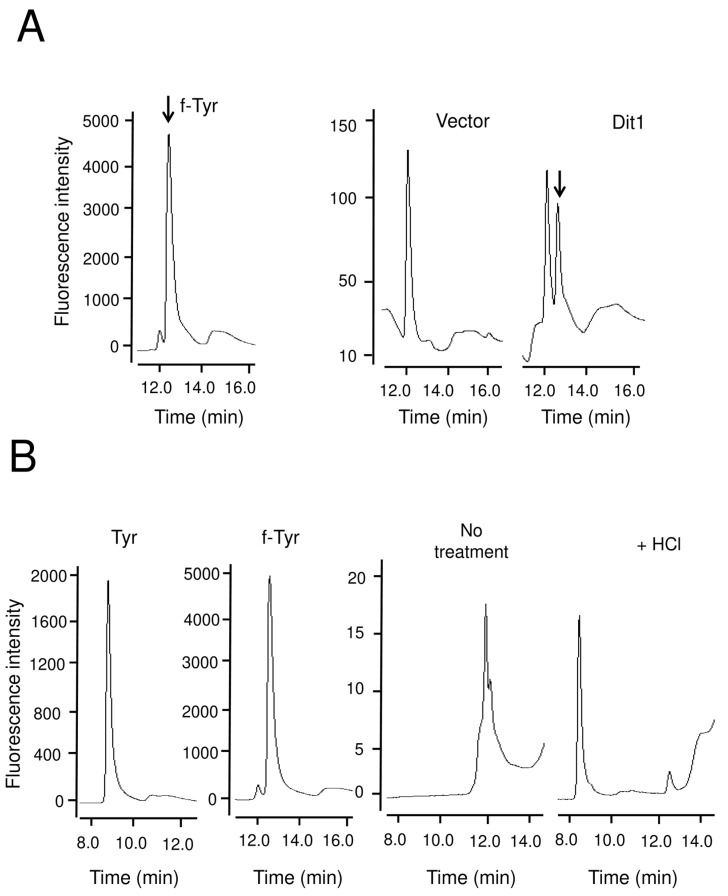
Trisodium citrate treatment of yeast lysates and formyl tyrosine detection. (**A**) The lysate of vegetative cells harboring pRS424GAL1-DIT1-GFP (Dit1) or pRS424GAL1 (Vector) were treated with trisodium citrate and subjected to HPLC analysis. Formyl tyrosine (f-Tyr) is shown as a control. Arrows indicate formyl tyrosine peaks. (**B**) The formyl tyrosine peak detected in the lysates of vegetative cells harboring pRS424GAL1-DIT1-GFP was collected, hydrolyzed with 6 N HCl and subjected to HPLC analysis. Tyrosine (Tyr) and formyl tyrosine (f-Tyr) are shown as controls.

**Figure 7 jof-06-00077-f007:**
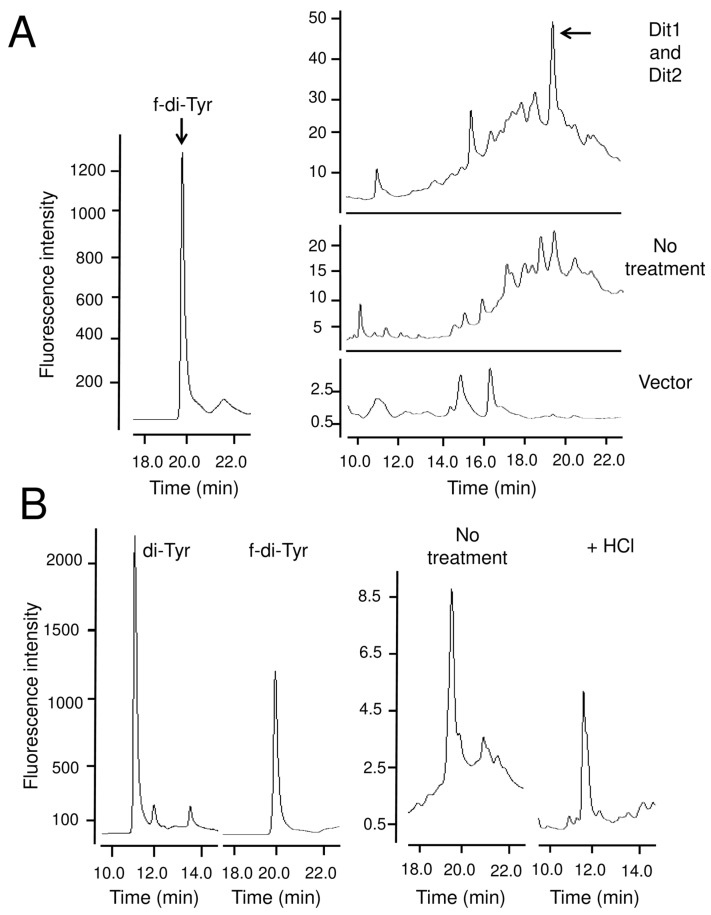
Bisformyl dityrosine detection in vegetative cells coexpressing Dit1 and Dit2. (**A**) The lysate of vegetative cells harboring pRS424GAL1-DIT1-GFP and pRS426GAL1-DIT2-FlAG (Dit1 and Dit2), or pRS424GAL1 and pRS426GAL1 (Vector) were treated with or without (no treatment) trisodium citrate and subjected to HPLC analysis. Formyl dityrosine (f-di-Tyr) is shown as a control. Arrows indicate formyl dityrosine peaks. (**B**) The formyl dityrosine peak detected in the lysates of vegetative cells harboring pRS424GAL1-DIT1-GFP and pRS426GAL1-DIT2-FlAG was collected, hydrolyzed with 6 N HCl and subjected to HPLC analysis. Dityrosine (di-Tyr) and formyl dityrosine (f-di-Tyr) are shown as controls.

**Table 1 jof-06-00077-t001:** Strains used in this study.

Strain	Genotype	Source
AN120	*MATα/MATa ARG4/arg4-NspI his3∆SK/his3∆SKho::LYS2/ho::LYS2* *leu2/leu2 lys2/lys2 RME1/rme1::LEU2trp1::hisG/trp1::hisG ura3/ura3*	[13]
AN117-4B	*MAT α ura3 leu2 trp1 his3∆SK arg4-NspI lys2 ho::LYS2 rme1::LEU2*	[13]
AN117-16D	*MATa ura3 leu2 trp1 his3∆SK lys2 ho::LYS2*	[13]
YPH499	*MATa ura3-52 lys2-801_amber ade2-101_ochre trp1-Δ63 his3-Δ200 leu2-Δ1*	[12]
HW3 (*dit1∆*)	*MATα/MATa ARG4/arg4-NspI his3∆SK/his3∆SK ho::LYS2/ho::LYS2 leu2/leu2 lys2/lys2RME1/rme1::LEU2 trp1::hisG/trp1::hisG ura3/ura3 dit1∆::his5+/dit1∆::his5+*	[14]
HI391	*MATα/MATa ARG4/arg4-NspI his3∆SK/his3∆SKho::LYS2/ho::LYS2* *leu2/leu2 lys2/lys2 RME1/rme1::LEU2trp1::hisG/trp1::hisG ura3/ura3* *DIT1::GFP-HIS3*	this study

**Table 2 jof-06-00077-t002:** Oligonucleotide primers used in this study.

Name	Sequence
HP128	GTGTAAGCTTATGGTGAGCAAGGGCGAGG
HP129	GTGTCTCGAGTTACTTGTACAGCTCGTCCA
HXO683	GTGTACTAGTAAAATGACATTTACTAGCAA
ECOR1-DIT1 -r	GTGTGAATTCAGAGATTTTCTTGATAACGA
HXO685	GTGTACTAGTCAAATGGAGTTGTTAAAGCT
DIT2-FLAG-r	CCGGGGGATCCTTACTTATCGTCGTCATCCTTGTAATCTCCTGCTCCTGCTTCCATTATATTCTCGTTAA
DIT1-GFP-F-INT	CTCGTTTCGATATTGGAGAAGGAGGACATTTCGTTATCAAGAAAATCTCTCGGATCCCCGGGTTAATTAA
DIT1-GFP-R-INT	TGTTTAAGTAAAAGAACAAAAAGGTAGACCAATGTAGCGCTCTTACTTTAGAATTCGAGCTCGTTTAAAC
DIT1-HA-r	GTGTGAATTCTTATGCATAATCCGGAACATCATACGGATATCCTGCTCCTGCAGAGATTTTCTTGATAACGA

**Table 3 jof-06-00077-t003:** Plasmids used in this study.

Name	Description	Source
pRS316TEF-PvcA	*URA3*; for expression of PvcA from The constitutive *TEF2* promoter	this study
pRS424GAL1pr-DIT1-GFP	*TRP1*; for expression of Dit1-GFP from the *GAL1* promoter	this study
pRS426GAL1pr-DIT2-FLAG	*URA3*; for expression of Dit2-FLAG from the *GAL1* promoter	this study
pRS424GAL1pr-DIT1-HA	*TRP1*; for expression of Dit1-HA from the *GAL1* promoter	this study

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
