# Peer review of "Studies on the Properties of the Sporulation Specific Protein Dit1 and Its Product Formyl Tyrosine"

_jof, 2020, doi:10.3390/jof6020077_

Round 1

Reviewer 1 Report

The authors present valuable data. The authors tried to demonstrate the functions of Dit1 by many methods. The experimental setup and methods used are well thought out.

I would ask the authors to include in the discussions future research perspectives and briefly characterize why these studies are so important?

I am also asking for information on whether transcriptome or proteome tests have been performed for the dit1 mutant, if so, please describe which biological or molecular functions are up or down regulated. It is also interesting what biological functions are affected by overexpression of this gene. If the authors have such data, please discuss them.

Line 191 - P. aeruginosa

Line 199 Authors do not present these data, please show these data in supplementary data. They also not comments these data. Additionally, what daes it mean 'sensitive' to CFW. In toxicology it mean that cells not growth, by cell wall disorder biosynthesis. Please change these term, cells are dye by CFW for my opinion

Author Response

(1) I would ask the authors to include in the discussions future research perspectives and briefly characterize why these studies are so important?

Response:

We have described future works (perspectives) including further characterization of formyl tyrosine- and formyl dityrosine-containing molecules and isolation of the donor substrate of Dit1. These have been included in the text (Page 15, Line 346-348 and Line 318-320). Additionally, we think formyl tyrosine and bisformyl tyrosine are intriguing as materials because the dityrosine layer is assembled on the chitosan and endows spores with unique properties. This statement has been added in Discussion (Page 15, Line 348-350).

(2) I am also asking for information on whether transcriptome or proteome tests have been performed for the dit1 mutant, if so, please describe which biological or molecular functions are up or down regulated. It is also interesting what biological functions are affected by overexpression of this gene. If the authors have such data, please discuss them.

Response:

Transcriptome data in dit1∆ cells or DIT1 overexpressing cells have not been reported. Additionally, effects of DIT1overexpression are not known. Thus, we have not modified the text regarding this comment.

(3)  Line 191 - P. aeruginosa

Response:

As the reviewer suggested, the text has been revised (Page 5, Line 191)

(4) Line 199 Authors do not present these data, please show these data in supplementary data. They also not comments these data. Additionally, what daes it mean 'sensitive' to CFW. In toxicology it mean that cells not growth, by cell wall disorder biosynthesis. Please change these term, cells are dye by CFW for my opinion

Response:

The reviewer may misunderstand our results. In Figure 1, we have shown that dityrosine layer formation is prevented by PvcA expression (Figure 1A, C and D). PvcA cannot complement DIT1 deletion (Figure 1B). Thus, we conclude that Dit1 is not an isocyanide synthase. These results and conclusions have been described in (Page 6, Line 197-207). To make a point clear that spores are stained with CFW by PvcA expression, percentage of cells stained with CFW in control cells (the cells harboring the empty vector) have been added in the text (Page 6, Line 201).

The reviewer pointed out that “CFW sensitive” is not appropriate. Regarding this suggestion, we have revised the sentence as “cells were stained with CFW” (Page 6, Line 200).

Reviewer 2 Report

In this manuscript, the authors have reported the properties of sporulation specific Dit1 protein and its product. This work would be useful for researchers studying yeast sporulation mechanisms in the future. There are few issues which needs attention:

  1. In the methods section (Line 114), the authors have mentioned that they used "5 mm diameter" glass beads. Is this true? or is it "0.5 mm" glass beads? Please clarify!
  2. Please mention the scale bar for all the microscope images.
  3. The x-axis labels are missing from the HPLC profiles

Author Response

(1) In the methods section (Line 114), the authors have mentioned that they used "5 mm diameter" glass beads. Is this true? or is it "0.5 mm" glass beads? Please clarify!

Response:

We used 0.5 mm beads. The text has been revised (Page 4, Line 114).

(2) Please mention the scale bar for all the microscope images.

Response:

As the reviewer suggested, scale bars have been added in microscopy images.

(3) The x-axis labels are missing from the HPLC profiles.

Response:

As the reviewer suggested, x-axis labels have been added in HPLC results (Figure 1, 3-7 and S2).